

# Predictors of one-year mortality following hip fracture surgery in elderly

Mehmet Özel[1], Mustafa Altıntaş[2] and Ali Cankut Tatlıparmak[3]

[1] Department of Emergency Medicine, University of Health Sciences, Diyarbakir Gazi Yasargil Training and Research Hospital, Diyarbakir, Turkey
[2] Department of Orthopedic Surgery, University of Health Sciences, Diyarbakir Gazi Yasargil Research and Training Hospital, Diyarbakir, Turkey
[3] Department of Emergency Medicine, Uskudar University, Faculty of Medicine, İstanbul, Turkey

## ABSTRACT

**Background**. Understanding mortality risk factors is critical to reducing mortality among elderly hip fracture patients. To investigate the effects of admission and post-operative levels of distribution width of red blood cells (RDW), albumin, and RDW/albumin (RA) ratio on predicting 1-year mortality following hip fracture surgery.

**Methods**. A retrospective study was conducted on 275 elderly patients who underwent hip fracture surgery in a tertiary hospital between January 2018 and January 2022. Deaths within one year of hip fracture were defined as the deceased group. The survivors were defined as those who survived for at least one year. The relationship between admission and post-operative levels of RDW, albumin, RA, and mortality within one year after hip surgery was assessed statistically, including binary logistic regression analysis. The study also assessed other factors related to mortality.

**Results**. One-year mortality was 34.7%. There was a 3.03-year (95% CI [1.32–4.75]) difference between the deceased ($79.55 \pm 8.36$ years) and survivors ($82.58 \pm 7.41$ years) ($p < 0.001$). In the deceased group, the mean hemoglobin (HGB) values at admission ($p = 0.022$) and post-operative ($p = 0.04$) were significantly lower. RDW values at admission ($p = 0.001$) and post-op ($p = 0.001$) were significantly lower in the survivor group. The mean albumin values at admission ($p < 0.001$) and post-operative ($p < 0.001$) in the survivor group were significantly higher than in the deceased group. A significant difference was found between the survivor group and the deceased group in terms of mean RA ratio at admission and post-operative ($p < 0.001$). Based on binary logistic regression analysis, presence of chronic obstructive pulmonary disease (COPD) (OR 3.73, 95% CI [1.8–7.76]), RDW (OR 1.78, 95% CI [1.48–2.14]), and albumin (OR 0.81, 95% CI [0.75–0.87]), values at admission were found to be independent predictors of 1-year mortality in elderly patients with hip fracture.

**Conclusion**. Based on this study, presence of COPD, higher RDW, and lower albumin levels at admission were independent predictors of 1-year mortality following hip fracture surgery in the elderly.

Corresponding author
Ali Cankut Tatlıparmak,
alicankut@gmail.com

## INTRODUCTION

Among the elderly, hip fractures are a serious morbidity and mortality risk (*Lin et al., 2018*; *Downey, Kelly & Quinlan, 2019*). It is uncommon for hip fractures to be fatal, but they are often associated with death. A hip fracture is a high-risk situation for an elderly patient, as the mortality rate in the first year after fracture is 12–35% (*Menéndez-Colino et al., 2018*). One-year mortality after hip fracture surgery was found to be 20% to 30% in a prospective, observational study of 2,660 patients (*Moran et al., 2005*). The reduction of mortality among hip fracture elderly patients depends on understanding mortality risk factors.

Elderly with hip fractures are particularly vulnerable to postoperative complications, postoperative morbidity, and mortality. This is due to a low level of preoperative physical activity and a variety of co-morbidities (*Kyriacou & Khan, 2021*). Performing clinical and laboratory evaluations before surgery will help identify patients at high mortality risk. In geriatric hip fractures, several factors may influence mortality. These factors include age, sex, American Society of Anesthesiologists (ASA) score, dementia grade, ability to walk, fracture type, surgical timing, type of surgery, length of hospitalization, and albumin levels (*Pimlott et al., 2011*; *Härstedt et al., 2015*; *Jiang et al., 2015*). Many clinical scoring systems such as the Charlson Comorbidity Index (CCI), Nottingham Hip Fracture Score (NHFS), especially the American Society of Anesthesiologists (ASA) score, have proven to be effective in divining hip fracture mortality rate (*Nelson, Scott & Sivalingam, 2020*; *Ek et al., 2022*).

Distribution width of red blood cells (RDW) is a measure of the volume and size range of red blood cells (erythrocytes). Complete blood counts (CBC) include RDW, which is readily available, inexpensive, and often performed as part of that protocol. Studies have investigated the role of RDW in predicting mortality and adverse outcomes in different medical conditions, including cardiovascular disease, diabetes, sepsis, trauma and burn injury (*Salvatori et al., 2019*; *Hamdan et al., 2021*; *Seo et al., 2022*; *Dankl et al., 2022*; *Marom et al., 2022*; *Hong et al., 2022*). Moreover, RDW is also used to predict the mortality of hip fractures alone or in combination with other factors (*Garbharran et al., 2013*; *Yin et al., 2016*; *Hamdan et al., 2021*; *Marom et al., 2022*). In numerous studies, low serum albumin levels have been found to negatively correlate with hip fracture mortality (*Pimlott et al., 2011*; *Laulund et al., 2012*; *Pass et al., 2022*).

The aim of this study is to investigate the potential effects of admission and postoperative levels of RDW, albumin, and RDW/albumin (RA) ratio on predicting 1-year mortality following hip fracture surgery in elderly patients. While individual associations of RDW and albumin with hip fracture mortality have been explored in prior studies, our research seeks to elucidate the combined effects and potential synergistic relationships between these biomarkers. These biomarkers have shown promise in previous research as potential indicators of mortality risk. By examining their roles in predicting outcomes, we seek to contribute to the understanding of mortality risk factors in this vulnerable population. The study's outcomes may fill crucial knowledge gaps and contribute to a comprehensive assessment of mortality risk factors in elderly hip fracture patients.

## MATERIALS & METHODS

### Study design, settings and patients

A 681-bed tertiary care hospital which was a level I trauma center, in Diyarbak ır, Southeastern Anatolia Region of Turkey. Servicing elderly patients who have fractures that require surgery in all districts of Diyarbakir and nearby provinces. An average of 300 elderly patients undergo fracture surgery each year at our hospital.

A retrospective single-centre study conducted between January 2018 and January 2022. We included patients 65 years of age or older (elderly) hospitalized in orthopedics and traumatology department from emergency department (ED) who were diagnosed with hip fractures. The study excluded elderly patients with hip fractures resulting from high energy trauma, such as motor vehicle accidents or firearm related injury, and pathological fractures, such as those caused by tumours.

In the event of hip trauma, patients are admitted to the emergency department for assessment and emergency medical intervention. X-rays or other imaging methods are used to confirm hip fracture diagnosis. The patient's general health status, comorbidities, medication use, allergies, laboratory results, and critical parameters such as electrocardiogram are evaluated. Patients are admitted to the hospital for hip surgery. Geriatric patients' cardiology and pulmonary status are examined by cardiologists and pulmonologists before surgery. Afterward, the anesthesiologist assesses the patient's suitability for anesthesia and decides the best method of anesthesia. The ASA scale is used by anesthesiologists to predict patient tolerance to anesthesia and surgery. The patient's preoperative blood levels are assessed, and a blood transfusion is ordered if necessary. The timing of the surgery is planned at a suitable time by the orthopedic surgeons. Patients are admitted to an intensive care unit after surgery. Postoperative pain control, blood transfusions if necessary, and other postoperative care are provided. Patients undergo postoperative rehabilitation. In order to regain mobility and function, physical therapists and rehabilitation specialists assist patients. Surgical wounds are monitored for infection, and necessary measures are taken to prevent infection. Following hip mobilization, patients are to be discharged.

### Clinical data collection and outcome assessments

Data on sociodemographics, including age, gender, ASA score and past medical histories, including hypertension, diabetes, cardiac disease, chronic obstructive pulmonary disease (COPD), chronic kidney disease, and malignancy were collected. Laboratory variables such as hemoglobin (HGB), hematocrit (HCT), RDW, and albumin were measured with the usual methods. All patients' laboratory data were collected on admission to ED and on their first postoperative day. To quantify the data, RDW was collected at the time of hospital admission and postoperatively, and albumin levels were measured using standardized laboratory techniques. The timing of RDW collection was crucial in capturing the initial disease severity and postoperative changes. The RA ratio was calculated for all patients as well. Radiographs performed in ED were analyzed to confirm the fracture type and identify the anatomic location in patients with hip trauma.

Among the variables analyzed were the need for blood transfusions, the length of hospital stay (LOS) (days), the time between hospitalization and surgery (in or out of office hours), the length of postoperative ICU follow-up (day), complications in the ICU (intubation, cardiopulmonary resuscitation, need for inotropes and vasopressors, multiple organ failure, presence of sepsis).

Deaths within the first year following fracture hospitalization were defined as 1-year mortality. The national database system was used to confirm whether the patients were alive or dead. Death causes were not collected. A survivor group and a deceased group were formed from all of the study patients. Patients who survived after hip fractures for at least one year were defined as survivor group. The deceased group was defined as patients who died within one year after a hip fracture.

## Ethics statement

Ethics committee approval and institutional permission in accordance with the Helsinki Declaration were obtained. Approval was granted by the Gazi Yasargil Training and Research Hospital Ethical Committee before the study (Decision Date: December 9, 2022 No: 268). Verbal consent was obtained from patients or their next of kin. Patients' personally identifiable information was encrypted, and all data analyzed were anonymous.

## Statistical analysis

The analysis of the data was performed using SPSS 28.0 for Mac (SPSS, Chicago, IL, USA). The objective of our statistical analysis was to examine the relationships between variables and identify the most relevant predictors of 1-year mortality following hip fracture surgery in the elderly population.

To build our study model, we followed a rigorous process that involved several steps. First, we conducted a thorough univariate analysis to explore the associations and differences between variables. Various statistical tests were employed, including the Fisher's exact test, chi-square ($\chi 2$) test, Student's $t$-test, and the Mann–Whitney U test. The primary aim of this analysis was to identify potential predictors that could be included in the subsequent multivariate analysis. In this analysis, we identified variables with $p$-values less than 0.2 as potential predictors to be included in the subsequent multivariate analysis. To compare the proportions of co-morbidities between the survivor group and deceased group, we employed both the Fisher's exact test and the chi-square ($\chi 2$) test. The Fisher's exact test was used when the expected cell counts were small or when the sample size was limited, ensuring accurate assessment of statistical significance in these cases. The chi-square test was used when the assumptions of the test were met, such as when the expected cell counts were reasonably large. It is important to note that for co-morbidities with relatively small sample sizes, including CKD, CHD, stroke, and cancer, the precision of estimates may be affected due to limited sample sizes. Nevertheless, we presented these results to provide an initial understanding of the distribution of these co-morbidities in our study population, while considering the limitations associated with small sample sizes. It is important to note that multiple hypothesis testing increases the risk of obtaining false positive results. One commonly used approach to address this concern is False Discovery

Rate (FDR) control (*Boca & Leek, 2018*). FDR control methods help mitigate the inflation of Type I errors when conducting multiple statistical tests. They are particularly valuable when a large number of parameters are being tested simultaneously. While FDR control is important, we opted not to employ it in our study due to the relatively limited number of parameters examined. Nevertheless, it is crucial to acknowledge that the inclusion of statistically significant variables in the multivariate model is not solely based on univariate analysis results. The final model is a comprehensive statistical framework that accounts for both statistical significance and clinical relevance.

For the multivariate analysis, we utilized binary logistic regression, which served as the core of our investigation. The logistic regression model allowed us to develop a predictive model that would enable us to understand the relationship between the independent variables and the dependent variable (1-year mortality). In building the logistic regression model, we employed the enter method, which involves the simultaneous inclusion of all relevant predictor variables in the model. This approach ensures that all potential predictors are considered and allows for a comprehensive evaluation of their individual effects on the outcome.

To ensure the robustness and validity of our model, we carefully assessed its performance and goodness of fit. We conducted tests for goodness of fit, including the Hosmer-Lemeshow test, to evaluate the adequacy of the model in predicting the outcome. Additionally, we assessed multicollinearity among the predictor variables to ensure that they were not highly correlated, which could affect the stability and interpretability of the model.

Furthermore, we accounted for potential confounding variables that were both statistically significant and recognized as important factors in previous studies, namely age, delay to surgery, and dementia (*Härstedt et al., 2015*; *Henderson & Ryan, 2015*; *Menéndez-Colino et al., 2018*). These variables were included in our logistic regression model to control for their potential influence on the relationship between the independent variables and the outcome. This approach allowed us to obtain a more accurate estimation of the independent effects of the predictors while considering the potential confounding factors.

## RESULTS

A total of 275 patients were included in the study. It was found that the mortality rate at 1 year was 34.7%. As shown on Table 1, the mean age in the deceased group (79.55 ± 8.36) was 3.03 (95% CI [1.32–4.75]) years lower than the survivor group (82.58 ± 7.41) ($p < 0.001$). There was no significant difference between sexes in 1-year mortality ($p = 0.303$). Hypertension among survivors ($n = 159$, 64.9%) was significantly lower than among the deceased ($n = 112$, 86.2%) ($p < 0.001$). Survivors (22.9% $n = 56$) had significantly lower rates of COPD than the deceased (50% $n = 65$) ($p < 0.001$). A significantly higher dementia and stroke rate was observed in the deceased group ($p = 0.004$ and $p = 0.006$, respectively). The location of the fracture ($p = 0.495$) and the side of the fractured extremity ($p = 0.086$) did not show statistically significant differences between the survivor and deceased groups. A statistical analysis of laboratory data (Table 2) found

**Table 1  Comparison of patient demographic data and co-morbid diseases by groups.**

|  |  | Survivor ($n = 245$) | Deceased ($n = 130$) | $p$ value |
|---|---|---|---|---|
| Age (year) |  | 79.55 ± 8.36 | 82.58 ± 7.41 | <0.001 |
| Sex | Female | 151 (61.6%) | 73 (56.2%) | 0.303 |
|  | Male | 94 (38.4%) | 57 (43.8%) |  |
| Co-morbidities | DM | 40 (16.3%) | 107 (17.7%) | 0.736 |
|  | CAD | 220 (89.8%) | 114 (87.7%) | 0.554 |
|  | HT | 159 (64.9%) | 112 (86.2%) | <0.001 |
|  | CKD | 2 (0.8%) | 3 (2.3%) | 0.229 |
|  | CHD | 8 (3.3%) | 10 (7.7%) | 0.056 |
|  | COPD | 56 (22.9%) | 65 (50%) | <0.001 |
|  | Dementia | 61 (24.9%) | 51 (39.2%) | 0.004 |
|  | Stroke | 10 (4.1%) | 15 (11.5%) | 0.006 |
|  | Cancer | 1 (0.4%) | 4 (3.1%) | 0.051 |
| Fracture location | Femoral neck | 82 (33.5%) | 36 (27.7%) | 0.495 |
|  | İntertrochanteric | 140 (57.1%) | 82 (63.1%) |  |
|  | Subtrochanteric | 23 (9.4%) | 12 (9.2%) |  |
| Fracture Side | Right | 115 (46.9%) | 49 (37.7%) | 0.086 |
|  | Left | 130 (53.1%) | 81 (62.3%) |  |

that the mean hemoglobin value at admission ($p = 0.022$) and the mean hemoglobin value 24 h after surgery ($p = 0.04$) were significantly lower in the deceased group. There was no significant difference between the groups in terms of hemoglobin change at admission and 24 h postoperatively ($p = 0.315$). RDW values at admission ($p = 0.001$) and post-operative ($p = 0.001$) were significantly lower in the survivor group, whereas RDW change ($p = 0.5$) was not significantly different between groups. The mean albumin value at admission ($p < 0.001$) and the mean albumin value after surgery ($p < 0.001$) in the survivor group were significantly higher than in the deceased group. A statistically significant difference between the admission and post-operative albumin values was not observed between the groups ($p = 0.181$). Mean RA ratio was significantly higher in the survivor group than in the deceased group at admission ($p < 0.001$) and post-op 24 h ($p < 0.001$), while the difference between admission and post-op was not statistically significantly different (0.961).

Blood transfusion was more frequent in the deceased group ($p = 0.021$). Surviving patients had surgery within 24 h at a significantly higher rate (69%, $n = 169$) than deceased patients (47.7%, $n = 62$) ($p < 0.001$). However, (Table 3) whether surgery was performed during or outside of office hours did not affect mortality at 1-year ($p = 0.923$). Postoperative intensive care follow-up lasted 1 (IQR 1-3) days in the survivor group and 3 (IQR 1-5) days in the deceased group, with a statistically significant difference between the two groups ($p = 0.001$). Median length of hospitalization was significantly lower in the survivor group (5 (IQR 3-7)) than in the deceased group (7 (IQR 4-11)) ($p < 0.001$). During follow-up, the deceased group ($n = 88$, 32.3%) had a significantly lower rate of sepsis ($n = 11$, 4.5%) than the survivors ($n = 11$, 4.5%) ($p < 0.001$). Survivors ($n = 164$, 66.9%) developed fewer

**Table 2  Comparison of laboratory data according to groups.**

| | Survivor ($n = 245$) | Deceased ($n = 130$) | $p$ value |
|---|---|---|---|
| Hemoglobin (admission) (g/L) | 12.3 (11.05–13.4) | 11.6 (10.53–13.3) | 0.022 |
| Hemoglobin (post-op) (g/L) | 10.2 (9–11.5) | 9.85 (9–10.88) | 0.04 |
| Hemoglobin difference (g/L) | 1.8 (1.2–2.6) | 1.6 (0.9–2.7) | 0.315 |
| RDW (admission) (%) | 13.7 (13.2–14.6) | 16.1 (14.3–17.43) | <0.001 |
| RDW (post-op) (%) | 13.7 (13.2–14.7) | 16 (14.38–17.43) | <0.001 |
| RDW difference (%) | 0.1 (−0.2–0.28) | 0.1 (−0.2–0.3) | 0.5 |
| Albumin (admission) (g/L) | 37 (32.13–40) | 30 (28–32) | <0.001 |
| Albumin (post-op) (g/L) | 34 (28–37) | 27 (24.75–29.93) | <0.001 |
| Albumin difference (g/L) | 3 (2–4) | 2 (2–4) | 0.181 |
| RDW-to-albumin ratio (admission) | 0.38 (0.34–0.45) | 0.55 (0.47–0.62) | <0.001 |
| RDW-to-albumin ratio (post-op) | 0.42 (0.37–0.51) | 0.59 (0.51–0.7) | <0.001 |
| RDW-to-albumin ratio difference | 0.92 (0.87–0.96) | 0.93 (0.86–0.96) | 0.961 |

**Table 3  Comparison of data on patient follow-up according to groups.**

| | Survivor ($n = 245$) | Deceased ($n = 130$) | $p$ value |
|---|---|---|---|
| Blood transfusion | 32 (13.1%) | 29 (22.3%) | 0.021 |
| ASA score | 3 (3–4) | 4 (3–4) | <0.001 |
| Surgery time (within 24 h) | 169 (69%) | 62 (47.7%) | <0.001 |
| Surgery time (>24 h) | 76 (31%) | 68 (52.3%) | <0.001 |
| Surgery period (Office hours) | 214 (87.3%) | 114 (87.7%) | 0.923 |
| Surgery period (Out-of-office hours) | 31 (12.7%) | 16 (12.3%) | |
| Post-operative ICU follow-up (days) | 1 (1–3) | 3 (1–5) | <0.001 |
| Lenght of hospital stay | 5 (3–7) | 7 (4–11) | <0.001 |
| No complication in ICU | 164 (66.9%) | 46 (22.3%) | <0.001 |
| Sepsis | 11 (4.5%) | 88 (32.3%) | <0.001 |

complications during intensive care follow-up than deceased patients (22.2% $n = 46$) ($p < 0.001$).

We performed a binary logistic regression analysis using the 'enter' method, which involved including all potentially relevant variables identified through the univariate analysis that showed statistical significance. We conducted tests to assess the goodness of fit in our logistic regression models. Specifically, we employed the Hosmer-Lemeshow goodness-of-fit test to evaluate the adequacy of our models. The test resulted in a chi-square value of 3.02 and a $p$-value of 0.933, indicating a good fit for our logistic regression model. Using Nagelkerke's R square test, the result was 0.550. The model performed well with an accuracy rate of 81.8%. The area under the curve of the model was 0.885 (95% CI [0.850–0.919])($p < 0.001$).

Based on the analysis, (Table 4) COPD presence (OR 3.73, 95% CI [1.8–7.76]), RDW value at presentation (OR 1.78, 95% CI [1.48–2.14]), and albumin value at presentation (OR 0.81, 95% CI [0.75–0. 87]), were found to be independent predictors of 1-year mortality in patients presenting with hip fracture to the emergency department.

**Table 4  Results of multivariate analysis.**

| Predictor | Reference level | *p* value | OR (95% CI) |
|---|---|---|---|
| COPD | No COPD | <0.001 | 3.73 (1.8–7.76) |
| RDW (at admission) | – | <0.001 | 1.78 (1.48–2.14) |
| Albumin (at admission) | – | <0.001 | 0.81 (0.75–0.87) |

Notes.
COPD, chronic obstructive pulmonary disease; RDW, distribution width of red blood cells; OR, Odds ratio; CI, Confidence interval.

# DISCUSSION

In this retrospective study, at admission, COPD presence, RDW, and albumin were strongly and independently associated with 1-year mortality in 275 elderly patients with hip fractures treated surgically in a tertiary hospital. Through multivariate adjustment, this study shows that, during admission, higher RDW levels (OR 1.78, 95% CI [1.48–2.14]), lower albumin levels (OR 0.81, 95% CI [0.75–0.87]), and presence of COPD were all independent predictors for 1-year mortality in the elderly.

A number of biochemical markers can be tested routinely before surgery in elderly patients with hip fractures, and they are relatively inexpensive and easy to interpret. As well, biochemical markers are modifiable factor that can improve outcomes and may be superior to non-modifiable factors like age, gender and fracture type in preventing mortality. RDW is included in CBC, which is readily available and inexpensive. According to several studies, high RDW levels are closely linked to mortality (*Laulund et al., 2012*, p. 6; *Lv et al., 2016*; *Yin et al., 2016*; *Pass et al., 2022*). A retrospective study of 1574 elderly patients who underwent hip fracture surgery was conducted. Mortality at 3 months, 6 months and 1 year following hip fracture surgery was significantly related to higher preoperative RDW values (*Marom et al., 2022*). In elderly people with hip fractures, a previous study reported that increased RDW was remarkably associated with short-term (in-hospital, 4-month) and long-term (1-year) mortality following hip surgery (*Garbharran et al., 2013*). Among 1479 hip fracture patients studied in a prospective cohort study, high RDW on admission was associated with long-term mortality (2-year and 4-year) (*Lv et al., 2016*). In this study, compared to the survivors group, RDW values measured both at admission (13.7% *vs* 16.1%) and postoperative (13.7% *vs* 16.0%) were found significantly higher in the deceased group. Also, a higher RDW at admission was independently associated with 1-year mortality in elderly patients. Based on our study, RDW appears to be a valuable biomarker for predicting long-term mortality in elderly people with hip fractures, consistent with literature.

The level of serum albumin is commonly considered a sign of protein depletion, and it is included in most metabolic assessments. In general, serum albumin measurements are relatively inexpensive and readily available. Low albumin levels have been associated with a lower morbidity and mortality rate in patients suffering from hip fractures in previous studies (*Pimlott et al., 2011*; *Laulund et al., 2012*; *Pass et al., 2022*). In a prospective study of 583 elderly patients with hip fractures, a statistically significant association (OR 2.44, 95% CI [1.17–5.12]) was found between low albumin levels at admission and mortality (*Pimlott et al., 2011*). Based on a retrospective analysis of 471 elderly hip fracture patients,

the median preoperative albumin level in those who died was 29.5 g/dl (SD 6.22 g/dl) and 32.8 g/dl (SD 6.43 g/dl) in those who survived. An association was found between preoperative albumin level and survival (hazard ratio 0.957, 95% CI [0.9377, 0.9978], $p < 0.001$) (*Harrison et al., 2017*). Another study in elderly patients with hip fractures found that low albumin levels (<29 g/L) were significantly associated with 30-day mortality following surgery (*Lizaur-Utrilla et al., 2019*). Our study differentiates itself by focusing on the combination of RDW, albumin parameters, and the RDW/albumin ratio as potential prognostic indicators of mortality in elderly hip fractures. According to our study, lower albumin levels at admission were a significant independent predictor of 1-year mortality following hip surgery in the elderly. Moreover, the deceased group had significantly lower albumin values both at admission (30 g/L *vs* 37 g/L) and postoperatively (27 g/L *vs* 34 g/L) than the survivors group. Our study findings supports the findings of previous studies and shows that albumin is a useful parameter for predicting mortality following hip fractures in elderly patients. Identifying and treating hypoalbuminemia early can be decrease mortality rates in elderly patients.

The combination of RDW and albumin (RA ratio) has been closely associated with mortality in certain diseases such as diabetes, burn, coronary artery disease, and sepsis patients (*Li, Ruan & Wu, 2022*; *Seo et al., 2022*; *Weng et al., 2022*; *Hong et al., 2022*; *Gu et al., 2022*). However, the association of RA ratio with mortality in elderly patients with hip fractures was unclear. In our study, both admission and post-operative RA ratio of the deceased group (0.55 and 0.59) was found remarkably higher than those of the survivor group (0.32 and 0.42). For the elderly, the RA ratio may serve as a new surrogate marker of increased mortality following hip fracture surgery.

Having several comorbidities and being older significantly increases the risk of mortality after hip fractures. Additionally, patients with comorbidities such as congestive heart failure, dementia, COPD, and malignancy have a 1.5–3 times higher 1-year mortality rate (*De Luise et al., 2008*). In a study conducted on elderly patients with hip fractures, COPD, and coronary artery disease were determined as independent risk factors for mortality (*Pimlott et al., 2011*). The 1-year mortality rate for 206 elderly patients with hip fractures was investigated in a prospective study. A remarkable association was found between those with respiratory complications and COPD. A total of 63% of those with respiratory complications also had COPD (*Henderson & Ryan, 2015*). In our study, comorbid conditions such as HT, COPD, dementia, and stroke were significantly higher in the deceased group compared to the survivors. Furthermore, 50% ($n = 65$) of the deceased patients had COPD. The presence of COPD at admission was found a significant independent predictor of 1-year mortality following hip surgery in the elderly. In addition to confirming previous findings, our findings strengthen the independent effect of COPD on mortality.

This study contributes to the understanding of mortality risk factors in elderly hip fracture patients and provides valuable insights into clinical practice. By identifying easily measurable and monitored predictors of mortality, this research equips healthcare professionals with valuable tools for managing and improving outcomes in this vulnerable population. Our study utilized a robust statistical analysis and involved a comprehensive

dataset comprising a significant number of elderly patients with hip fractures. The inclusion of multiple potential predictor variables enhances the reliability and applicability of our findings. Additionally, the study's conduct in a tertiary care hospital, which performs a high volume of hip fracture surgeries, strengthens the generalizability of our results. These strengths, along with the consistent associations found between RDW, albumin, and mortality in various medical conditions, support and reinforce existing knowledge in the field.

## LIMITATIONS

One of the study's limitations is that it was conducted retrospectively, at a single center. Second, the cause of death for the elderly population could provide insight into the factors causing such a high mortality rate. However, the causes of death are recorded in Turkish death certificates, but they weren't available for this study. Researchers should evaluate the association between RDW, albumin, RA ratios, and mortality, prospectively with large patient groups and in multiple centers, including causes of death in the elderly.

In this study, strict adherence to the predefined protocol for patient enrollment and data collection was prioritized. However, it is important to acknowledge that, despite our best efforts, some deviations from the protocol may have occurred. These deviations were generally minimal and can be attributed to the inherent challenges associated with retrospective study designs and reliance on existing medical records. The retrospective nature of our study introduced certain limitations, as it relied on the availability and completeness of medical documentation. Variations in the timing of data availability and potential differences in the completeness of certain variables could have arisen due to diverse documentation practices among healthcare providers and the availability of historical records. It is crucial to recognize that such deviations were unintentional and beyond the control of the research team.

## CONCLUSIONS

This study reveals that at admission higher RDW, lower albumin levels, and presence of COPD were found as independent predictors of 1-year mortality following hip fracture surgery in the elderly. Utilizing objective, readily available parameters such as RDW, albumin, and RA ratio can be effective in predicting and preventing mortality in elderly.

### Funding
The authors received no funding for this work.

### Competing Interests
The authors declare there are no competing interests.

## Author Contributions

- Mehmet Özel conceived and designed the experiments, performed the experiments, analyzed the data, prepared figures and/or tables, authored or reviewed drafts of the article, and approved the final draft.
- Mustafa Altıntaş conceived and designed the experiments, performed the experiments, authored or reviewed drafts of the article, and approved the final draft.
- Ali Cankut Tatlıparmak analyzed the data, prepared figures and/or tables, authored or reviewed drafts of the article, and approved the final draft.

## Human Ethics

The following information was supplied relating to ethical approvals (*i.e.*, approving body and any reference numbers):

Diyarbakir Gazi Yasargil Training and Research Hospital Ethical Board approved the study (Decision Date: December 9, 2022 No: 268)

## Data Availability

The raw data is available in the Supplemental File.

## Supplemental Information

Supplemental information for this article can be found online at http://dx.doi.org/10.7717/peerj.16008#supplemental-information.

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
