# Peer review of "Predictors of one-year mortality following hip fracture surgery in elderly"

_PeerJ, doi:10.7717/peerj.16008_

## Round 0.1 · original submission · Major Revisions

The main considerations are related to the state of the art that supports the research problem, the proposed statistical analysis (eg, greater amount of detail and clarity of the analysis), and correct interpretation of the results (study design does not explain cause-effect). For these reasons, the report needs extensive review and reassessment by the reviewers. For more details, please review the reviewers' comments.

·

Basic reporting

The article is clear and precise. Their background manages to explain the problem to be investigated. I have some doubts that I will comment on in the following sections.

Experimental design

In the methods, I have some comments:

1. Although some aspects of the care hospital are detailed, it would be useful to delve into the pre and post-surgery protocols to understand the context of the individuals studied. For example, a study with similar characteristics points out the following: "In the involved hospitals, early surgery (within 48 hours from the trauma) has been guaranteed since the early 2000’s. After arriving at the ED, the patient is triaged, prioritized by the admissions nurse and transferred to an orthopedic surgery department or to an orthogeriatric department, depending on beds’ availability. The duty orthopedic surgeon carries out a physical examination and assessment and then establishes the surgical technique. The inpatient rehabilitation treatment, aiming at allowing an early verticalization and walking, is started the day after surgery. It consists of two physical therapy sessions a day for six days a week. After the postoperative hospitalization phase a different pathway is defined for each patient tailored to their rehabilitation and nursing needs. This pathway might involve discharge from hospital to a rehabilitation care home, nursing home, or home care. The choice of most appropriate setting and intensity of treatment is made by an internal multi-professional team based on several elements, such as the patient’s clinical condition, cognitive status, social care network and orthopedic indication for weight bearing (early or delayed). During the period of the study, no changes in the care regime of hip-fracture patients were implemented." https://link.springer.com/content/pdf/10.1038/s41598-019-55196-6.pdf

2. I suggest describing the statistical analysis in more detail. Detail what was the objective of each statistical test. Also, indicate if there were confounding variables in the logistic regression analysis and if one or several regression models were obtained.

3. Declare the regression method used (for example, forward, backward, or other) and indicate that tests were performed for the goodness of fit in logistic regression models.

Validity of the findings

The novelty of this research lies in the analysis of variables such as RDW, albumin, and RDW/Albumin (RA) ratio. Statistical analysis is adequate, but as mentioned above it must be clearly described, since it is crucial to understand the results presented.

In the discussion, it would be appreciated if you add some strengths of your research.

Additional comments

NO COMMENTS

Reviewer 2 ·

Basic reporting

The objective of the study were clearly stated. The authors provided a clear and comprehensive description of the study design outlined the recruitment process and inclusion criteria for participants, and details of the collection process of each variable are provided.

The introduction section need more information. I would expect a little detailing of techniques used to quantify the data, such as the timing of RDW was collected. And how the current study different from other studies.

Experimental design

In the statistical analysis section, more details need to be added to explain the process of building and validating the proposed model. It is important to provide a clear and comprehensive description of the steps taken to build the study.

Also, the author may want to consider add more discussion about the protocol deviation and add more information about the process of enrollments.

Validity of the findings

Table 1, the results shows about the correlation which is not necessary the causation. The result interpretation need to be justified on line 136 where the author mentioned the one -year mortality was not affected by the location of fracture or side of fracture, which is not appropriate.

The author should add more discussion about the multiple testing issue when hypothesis testing in the univariate logistic regression analyses were done, there are multiple literature could be added for FDR control.

The RDW was collected in percentage, the author may want to consider using total number of cell count as predictors.

Also, with the multivariate model results in table 4, the author should add the reference level about the OR for interpretation.

Reviewer 3 ·

Basic reporting

In section “statistical analysis”, the authors mentioned that “binary logistic regression was used to verify the results of the univariate analysis. Statistical significance was determined by an alpha value of 0.05 (in the multivariate analysis).” Did the authors build logistic regression for both univariate analysis and multivariate analysis? If so, please present the results including OR and p-value for univariate logistic regression models since no table was presented in this paper to show the results. If not, please rephrase this paragraph to make it clear.

Experimental design

This study conducted a retrospective study on 275 elderly patients who underwent hip fracture surgery. The sample size seems large enough. The study design makes sense.

Validity of the findings

The conclusion from this study that presence of COPD, higher RDW, and lower albumin levels at admission were independent predictors of 1-year mortality following hip fracture surgery in elderly is impressive. However, I have main concerns on statistical analysis. These have to be clarified to prove that the results and conclusion are promising.

Please clarify the statistical method the authors used for two proportion comparison in co-morbidities. In Table 1, each co-morbidity has a p-value. Did you use two proportion Z-test? Please take care to use statistical test for the comparison in two proportions since most of existing methods have normality and large sample size assumption. Let’s take CKD, CHD, Stroke, and Cancer for example, the sample size for survivor group and deceased group with each of these co-morbidities too small to satisfied the assumption.

For multivariate logistic regression model, did the authors adjust confounders, e.g. Age, Sex, etc.? Covariate adjustment should be added to the logistic regression model. Also, which covariates were adjusted in the model should be justified.

Additional comments

NA

---

## Round 0.2 · Minor Revisions

Dear authors. Although they have produced a satisfactory response letter to the comments of this editor and their reviewers, it lacks the necessary information to be able to match/identify a response to the reviewer's observation with the modification made to the manuscript. I suggest that you reconstruct the response letter including the lines where the modification was made in the manuscript. On the other hand, in the introduction and methods sections there are sentences/sentences that are not specific to these sections. For example, appreciations or interpretations of results are indicated:

Introduction
To quantify the data, RDW was collected at the time of hospital admission and postoperatively, and albumin levels were measured using standardized laboratory techniques. The timing of RDW collection was crucial in capturing the initial disease severity and postoperative changes. This study stands apart from previous research by providing detailed information on the specific techniques used to quantify the data, ensuring accurate and reliable measurements. Furthermore, our study differentiates itself by focusing on the combination of RDW, albumin parameters, and the RDW/Albumin ratio as potential prognostic indicators of mortality in elderly hip fractures....

Clinical data collection and outcome assessments
In this study, strict adherence to the predefined protocol for patient enrollment and data collection was prioritized. However, it is important to acknowledge that, despite our best efforts, some deviations from the protocol may have occurred. These deviations were generally minimal and can be attributed to the inherent challenges associated with retrospective study designs and reliance on existing medical records. The retrospective nature of our study introduced certain limitations, as it relied on the availability and completeness of medical documentation. Variations in the timing of data availability and potential differences in the completeness of certain variables could have arisen due to diverse documentation practices among healthcare providers and the availability of historical records. It is crucial to recognize that such deviations were unintentional and beyond the control of the research team....

All these aspects negatively affect the basic reporting of the investigation. Review these aspects in order to make an editorial decision.

·

Basic reporting

The article is clear and precise. Their background manages to explain the problem to be investigated.

Experimental design

no comment

Validity of the findings

no comment

Additional comments

The authors have resolved the observations made in the first revision.

Reviewer 2 ·

Basic reporting

Thanks for addressing all my comments.

Experimental design

no comment any more

Validity of the findings

The author addressed all my comments.

Reviewer 3 ·

Basic reporting

The authors have added details in statistical method part and clarify the reason why the specific statistical methods were used. Now the statistical method part is clear. Besides, the authors used clear and unambiguous English. The literature references have been refined. The article structure, figures and tables look good.

Experimental design

The study design makes sense since the study design makes sense. This study conducted a retrospective study on 275 elderly patients who underwent hip fracture surgery. The experimental design of this study has also been refined based on the other reviews' comments.

Validity of the findings

Since main concerns in statistical methods have been solved, the results and conclusions derived from this study are relatively promising. Meanwhile, the underlying data have been provided which are robust. The conclusions are clearly stated and closely related to the research questions.

Additional comments

I recommend to accept the refined paper.

---

## Round 0.3 · accepted · Accept

The authors have addressed all reviewer and editor comments on the previous version. In addition, the authors have made changes related to the form and presentation of the research report. The current version of the manuscript is accepted.